## Reply

statistics

**Author for correspondence:**
David Colquhoun
e-mail: d.colquhoun@ucl.ac.uk

# A response to critiques of 'The reproducibility of research and the misinterpretation of *p*-values'

David Colquhoun

NPP, UCL, Medical Sciences, UCL, Gower Street, London WC1E 6BT, UK

(iD) DC, 0000-0002-4263-017X

## 1. Introduction

I proposed [1–3] that *p*-values should be supplemented by an estimate of the false positive risk (FPR). FPR was defined as the probability that, if you claim that there is a real effect on the basis of *p*-value from a single unbiased experiment that you will be mistaken and the result has occurred by chance. This is a Bayesian quantity and it means that there is an infinitude of ways to calculate it. My choice of a way to estimate FPR was, therefore, arbitrary. I maintain that it is a reasonable way, and has the advantage of being mathematically simpler than other proposals and easier to understand than other methods. This might make it more easily accepted by users. As always, not every statistician agrees. This paper is a response to a critique of my 2017 paper [2] by Arandjelović [4].

## 2. First some minor matters

In his critique of my 2017 paper [2], Arandjelović says [4, p. 1] that my argument 'provides little if any justification for the continued use of the *p*-value'. I agree because I never contended that it did. I recommended [3, p. 198] that authors should

> In my proposal, the terms 'significant' and 'nonsignificant' would not be used at all. This change has been suggested by many other people, but these suggestions have had little or no effect on practice. A *p*-value and confidence interval would still be stated but they would be supplemented by a single extra number that gives a better measure of the strength of the evidence than is provided by the *p*-value.

Furthermore, calculation of the *p*-value is the simplest way to calculate what I believe really matters, the FPR. It is an input to our Web calculator [5].

He also says [4, p. 1] 'Although criticisms [of *p*-values] are not new, until recently they were largely confined to the niche of the statistical community'. Actually criticisms have been voiced by

users for a long time as exemplified by the opening quotation in my paper. Baken, writing in the *Psychological Bulletin* in 1966 [6], described the conventional null-hypothesis testing as an emperor who had no clothes. I would agree that these early warnings have had little effect on practice, but the reason for that is not primarily a statistical one, but rather a consequence of the perverse incentives that are imposed on both authors and journal editors to produce discoveries without worrying too much about whether the results are just chance.

## 3. The major criticism

The main criticism of my piece in [4] seems to be that my calculations rely on testing a point null hypothesis, i.e. the hypothesis that the true effect size is zero. He objects to my contention that the true effect size can be zero, 'just give the same pill to both groups', on the grounds that two pills cannot be exactly identical. He then says 'I understand that this criticism may come across as frivolous semantic pedantry of no practical consequence: of course that the author meant to say "pills with the same contents" as everybody would have understood' [4, p. 2]. Yes, that is precisely how it comes across to me. I shall try to explain in more detail why I think that this criticism has little substance.

Nevertheless, I am grateful to Arandjelović [4] because his invective has given me the chance to discuss in a bit more detail the assumptions behind my views. Many of his criticisms have already been answered in my 2019 paper [3], which has been online in arXiv since 2018.

One puzzling aspect of his criticisms is that they do not mention the crucial matter of likelihood ratios.

## 4. Likelihood ratios

In my 2017 paper [2], I suggested three different ways of expressing uncertainty that improved on giving *p*-values alone. One was to use the reverse Bayes approach to calculate the prior probability that you would need to believe in order to achieve a specified FPR. This overcomes the fact that that we usually have no information about the prior probability. But it has the disadvantage that it risks creating a 'bright line' at FPR = 0.05, and that sort of dichotomization is universally deplored [7]. It also has the disadvantage that the idea of a prior probability is slippery and unfamiliar to most users.

As an alternative approach, I suggested giving the likelihood ratio, or its equivalent FPR based on the assumption that the prior probability is 0.5. This is now my first choice [3]. The Bayes' factor is the bit of Bayes' theorem that represents the strength of the evidence from the experiment, so it is the most natural thing to use [8]. Under my assumptions [3], the Bayes' factor becomes the likelihood ratio. It measures how probable your observations would be under the hypothesis that there is a real effect, relative to how probable they would be under the null hypothesis of zero effect. Call this ratio $L_{10}$ (the subscript indicates that the ratio is defined for $H_1$, relative to $H_0$). The biggest value that this can take is that when the alternative hypothesis has its most likely value, the observed mean effect size. It turns out that even this maximum evidence in favour of the alternative does not reject the null hypothesis as often as the traditional *p*-value. It is well known that if you observe $p = 0.05$, the likelihood ratio in favour of the alternative hypothesis is only around 3 (e.g. [8] and §10 in [1]). (There are many ways to define the likelihood ratio: some are summarized in table 3 of Held & Ott [9] which shows that most methods imply a value of the order of 3.) Odds of 3 to 1 on their being a real effect are far less than the odds of 19:1 which might, mistakenly, be inferred from $p = 0.05$.

The use of likelihood ratios avoids altogether the arguments about Bayes' theorem. It is an entirely frequentist approach, and it is sufficient alone to show that the *p*-value, as it is often mistakenly interpreted, exaggerates the evidence against the null hypothesis.

Note that the likelihood ratio approach does not assert that the true effect size is exactly zero. It merely says that if $L_{10}$ is small, then the evidence that there is a real effect is weak. From this point of view, it is irrelevant that the null hypothesis is that the true effect size is *exactly* zero. In any case, a narrow distribution around zero gives much the same results [9,10]. I agree with Berger & Sellke [10] when they say 'for a large number of problems testing a point null hypothesis is a good approximation to the actual problem' [10, p. 114]. If the likelihood ratio is not sufficiently big in favour of a real effect to conclude that a real effect is unlikely, then one obviously does not conclude that the effect size is exactly zero.

It is important to realize that there are many different ways in which the likelihood ratio can be calculated [9]. In most discussions this is ignored, or at least not stated explicitly, and this has led to confusion. One crucial distinction between them is old. Lindley [11] pointed it out:

> In fact, the paradox arises because the significance level argument is based on the area under a curve and the Bayesian argument is based on the ordinate of the curve.
>
> —D.V. Lindley [11, p. 190]

This is illustrated in fig. 1 of [2]. Astonishingly, there seem to be no generally accepted names for the two approaches. I called them the *p-equals* approach and the *p-less-than* approach, and the distinction between them is discussed in detail in §3 of my 2017 paper [2] so I will not repeat it here. Most people (e.g. [12,13]) still use the latter approach, though the former is clearly appropriate for answering my question: how should you interpret the *p*-value from a single unbiased experiment. For any given *p*-value, the *p-equals* approach gives smaller likelihood ratio, or bigger FPR, than the *p-less-than* approach (fig. 2 in [2]). Neglect of this distinction has led some people to underestimate the risk of false positives.

One problem with advocating likelihood ratios as a measure of evidence is that they are unfamiliar with most users, and that there is no generally accepted criterion for how big they must be for you to have reasonable confidence that your results are not a result of chance alone. That is why I suggested that, rather than specifying the result as $L_{10}$, the odds should be expressed as a probability:

$$\text{FPR}_{50} = \frac{1}{1 + L_{10}}.$$

This can be interpreted, using Bayes' theorem, as the FPR when the prior odds are 1, i.e. when the prior probability of a real effect is 0.5 (e.g. see eqn (A6) in [3]). The advantage of doing this is that the FPR measures what most people still mistakenly think the *p*-value does. That makes it very easy for non-statisticians to understand. But the disadvantage is that it involves Bayes' theorem and that always means an outbreak of the statistics wars.

Insofar as the $\text{FPR}_{50}$ is just a transformation of $L_{10}$, it is entirely frequentist, but its interpretation as a posterior probability is not.

## 5. How much do *p*-values (as commonly misinterpreted) exaggerate the evidence against the null hypothesis?

Much of the argument in this area centres on whether or not *p*-values (as commonly misinterpreted) exaggerate the strength of the evidence against the null hypothesis. Most people think that they do, though the extent of the exaggeration depends on the precise assumptions that you make (e.g. [9,10]).

Everyone agrees that the *p*-value is not the probability that your results occur by chance alone, despite that being the most common interpretation placed on it by users. The fact that so many people believe this misinterpretation suggests that what they want to know is the probability that their results are owing to chance alone, so we need a proper definition of that term. In my 2017 paper [2, p. 2], I define it this way:

> What you want to know is when a statistical test of significance comes out positive, what the probability is that you have a false positive, i.e. there is no real effect and the results have occurred by chance. This probability is defined here as the false positive risk (FPR).

One way of looking at the difference between the *p*-value and the FPR is to note that they have different denominators. The numerator for both is the number of false positives in hypothetical replications of the experiment. If the criterion for a positive result is $p < 0.05$, then this will be 5% of all tests in which the null hypothesis is true. The *p*-value is the ratio of this number of false positives to the total number of tests in which the null hypothesis is true (which is not usually known). The FPR is the ratio of the number of false positives to the total number of positive tests, both false positives and true positives. Under realistic conditions, the former denominator is larger, so the *p*-value is smaller than the FPR. A numerical example is given at 26:00 in [14].

Another way to look at the difference between *p*-value and FPR is to look at confusion between them as an example of the error of the transposed conditional [15]. They measure quite different things so, in principle, they cannot be equal.

If we accept that what we want to know is the FPR, the question arises of how to calculate it, and that is where the problems begin. From a Bayesian point of view, the FPR is the posterior probability that the null hypothesis is true, $P(H_0 | \text{data})$. There are differences of opinion about how it should be calculated. Because it is a Bayesian concept, there is literally an infinitude of ways in which it can be calculated. Held & Ott [9] have reviewed the many possibilities. Which way should we choose?

As Senn [16] has pointed out, the disagreements about how to calculate FPR are essentially disagreements between different versions of the Bayesian argument, rather than a disagreement between frequentists and Bayesians. Frequentists have no way to calculate FPR.

Some people (e.g. Casella & Berger [17]) have argued that putting a lump of prior probability on the null hypothesis gives the null an unfair advantage over other possibilities. This is a matter of opinion. It seems quite fair to me. Casella & Berger [18, p. 344] say

> Their main thesis is that the frequentist $p$-value overstates the evidence against the null hypothesis although the Bayesian posterior probability of the null hypothesis is a more sensible measure.

> The large posterior probability of $H_0$ that Berger and Delampady compute is a result of the large prior probability they assign to $H_0$, a prior probability that is much larger than is reasonable for most problems in which point null tests are used.

> In fact, it is not the case that $p$-values are too small, but rather that Bayes point null posterior probabilities are much too big!

> Most researchers would not put a 50% prior probability on Ho. The purpose of an experiment is often to disprove Ho and researchers are not performing experiments that they believe, *a priori*, will fail half the time! We would be surprised if most researchers would place even a 10% prior probability on Ho.

Casella & Berger [18] seem to have much more faith in the ability of experimenters to guess the outcome of an experiment than I think is appropriate. Most bright ideas turn out to be wrong, so I would guess that a prior probability of 0.5 there being a real effect is often optimistic, rather than being much too low. In my analysis, observation of $p = 0.05$ would imply a prior, $P(H_1)$ of 0.87 to make the FPR the same as the $p$-value [5]. They contend that this is reasonable. I think that if you were to submit a paper that claimed you had made a discovery and that a necessary assumption for that claim to be true was that you were almost (90%) certain that the claimed effect was real before you did the experiment, your paper would be unlikely be accepted. Casella & Berger seem to think that it is legitimate to adjust your prior in order to make the FPR more or less the same as the $p$-value. This makes no sense at all to me.

I chose to use a point null hypothesis as prior and to use a simple alternative hypothesis [3,9]. This makes sense because it is exactly what you do when you simulate repeated $t$-tests, as in Colquhoun [1]. The rest of the prior probability is on the alternative hypothesis, and when this is given its most likely value, the observed mean effect size, we find that the likelihood ratio is, at most, about 3, as above.

At the other extreme, Senn [16] has shown that a prior can be chosen (for one-sided tests) that makes the $p$-value essentially identical with the FPR. But because the FPR and the $p$-value measure quite different things, there is no earthly reason why they should be the same. This seems about as sensible as saying that you can always choose a prior probability that makes the $p$-value the same as the FPR. Without hard evidence about the accuracy of the priors that are assumed these are mere parlour tricks.

The use of a simple alternative hypothesis is not crucial for my results. Other approaches, which test a point null hypothesis, give similar results, as shown in [3]. In particular, the approaches of Sellke *et al.*, [19], and of Johnson [20] give results that are quite close to mine, using priors for the alternative hypothesis that are designed to maximize the odds in favour of rejection of the null hypothesis, $H_0$. It turns out that they reject the null hypothesis much less often than the $p$-value. These conclusions strengthen still further the view that the $p$-values, as commonly misinterpreted, exaggerate the strength of evidence against the null hypothesis. The FPR, under any realistic assumptions, is bigger than the $p$-value and this must, to some extent, contribute to the reproducibility crisis.

## 6. A similar suggestion

Benjamin & Berger [21] in the recent series, *Moving to a World Beyond 'p < 0.05'* make a suggestion very similar to mine. They suggest that the $p$-value should be supplemented with the maximum Bayes' factor given by their approximation

$$\mathrm{BF}_{10} = \frac{1}{-ep\ln(p)},$$

where $e$ is the base of natural logarithms and $p$ is the observed $p$-value. If you have observed $p=0.05$, this is 2.46, again of the order of 3, though this method predicts somewhat higher false positive risks than most others (see fig. 3 and table 1 in [3]). For example, if we observed $p = 0.005$, my method implies $\mathrm{FPR}_{50} = 0.034$, whereas the Sellke & Berger limit implies 0.067 (table 1 in [3]).

## 7. Conclusion

It is true that putting a lump of prior probability on the null hypothesis gives a bigger FPR (for any given $p$-value) than many other choices of prior: it is a sceptical prior of the sort that is appropriate, e.g. for drug

regulators [22, p. 412]. In that sense, my values should perhaps be viewed as maximum FPRs. But in a different sense, they are minimum FPRs, because prior probabilities of a real effect may well be lower than the default value of 0.5.

The fact that the Bayesian approach with a lump of prior probability on the null hypothesis gives results that are identical with simply counting false positives in repeated simulations of $t$-tests is another reason to think that this approach is reasonable [1].

I would be quite happy for people to report, along with the $p$-value and confidence intervals, the likelihood ratio, $L_{10}$, that gives the odds in favour of there being a real effect, relative to there being no true effect. That is a frequentist measure and it measures the evidence that is provided by the experiment.

If these odds are expressed as a probability, rather than as odds, we could cite, rather than $L_{10}$, the corresponding probability $1/(1 + L_{10})$. I suggest that a sensible notation for this probability is $FPR_{50}$, because it can, in Bayesian context, be interpreted as the FPR when you assume a prior probability of 0.5. But because it depends only on the likelihood ratio, there is no necessity to interpret it in that way, and it would save a lot of argument if one did not.

I think that the question boils down to a choice—do you prefer an 'exact' calculation of something that cannot answer your question (the $p$-value), or a rough estimate of something that can answer your question (the FPR). I prefer the latter.

Data accessibility. This article does not contain any additional data.
Competing interests. I declare I have no competing interests.
Funding. I received no funding for this study.

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
