## [Reviewer comments · Royal Society Open Science]

Review History

RSOS-190819.R0 (Original submission)

Review form: Reviewer 1 (Stephen Senn)

Is the manuscript scientifically sound in its present form?

Yes

Are the interpretations and conclusions justified by the results?

Yes

Is the language acceptable?

No

Is it clear how to access all supporting data?

Yes

Do you have any ethical concerns with this paper?

No

Have you any concerns about statistical analyses in this paper?

No

Recommendation?

Accept with minor revision (please list in comments)

Comments to the Author(s)

1) You should give FPR in full and give a brief definition when first introducing it. (I appreciate that this is a reply to a comment on a paper of yours. However, not all readers will have the previous material fresh in their minds.)

2) Your use of likelihood ratio is non-standard. This is typically used to mean the ratio of the maximum likelihood under H_1 to that under H_0 . Not the ratio of the integrated likelihood.

3) I was puzzled at to why the symbol L_{10} was chosen. Why 10 I wondered. Then it occurred to me that it is 1 for H_1 and 0 for H_0 . I then realised that Johnson had used BF_{10} for the Bayes Factor. I think that it would be best to be explicit..

4) The agreement with Johnson is hardly surprising since that also assumes a lump of probability on the null hypothesis. You seem however, to be claiming it as some sort of truly independent validation. (Of course it validates the algebra at least to some degree.) I think you ought to recognise this somewhere.

Review form: Reviewer 2 (Leonhard Held)

Is the manuscript scientifically sound in its present form?

Yes

Are the interpretations and conclusions justified by the results?

Yes

Is the language acceptable?

Yes

Is it clear how to access all supporting data?

Not Applicable

Do you have any ethical concerns with this paper?

No

Have you any concerns about statistical analyses in this paper?

No

Recommendation?

Accept with minor revision (please list in comments)

Comments to the Author(s)

I mostly agree with this response and I believe it deserves to be published. I only want to add a few references that might improve the manuscript. I leave it to the author's discretion whether to include those.

Berger & Sellke (1987, JASA) discuss in Section 4.2 that deviations from point null hypotheses to small sets about H_0 basically give the same results on the corresponding Bayes factors and likelihood ratios. This is reviewed in more detail in Held and Ott (2018), Section 5.2, and would perhaps add to the current discussion.

From a scientific perspective, a related discussion of the suitability of point null hypotheses has appeared in response to the paper by Amrhein and Greenland ("Retire Statistical Significance", 2019, Nature). In particular the letters by Ioannidis ("Retiring statistical significance would give bias a free pass"), Johnson ("Raise the bar rather than retire significance") and Haaf et al ("Retire significance, but still test hypotheses") seem to be relevant in this context. I think the response would be even stronger if it would review and comment on this discussion.

Decision letter (RSOS-190819.R0)

02-Oct-2019

Dear Dr Colquhoun,

On behalf of the Editors, I am pleased to inform you that your Manuscript RSOS-190819 entitled "A response to O. Arandjelovic's critique of "The reproducibility of research and the misinterpretation of p -values"" has been accepted for publication in Royal Society Open Science subject to minor revision in accordance with the referee suggestions. Please find the referees' comments at the end of this email.

The reviewers and handling editors have recommended publication, but also suggest some minor revisions to your manuscript. Therefore, I invite you to respond to the comments and revise your manuscript.

- Ethics statement

- Data accessibility

<http://datadryad.org/submit?journalID=RSOS&manu=RSOS-190819>

- **Competing interests**

- **Authors' contributions**

- **Acknowledgements**

- **Funding statement**

Because the schedule for publication is very tight, it is a condition of publication that you submit the revised version of your manuscript before 11-Oct-2019. Please note that the revision deadline will expire at 00.00am on this date. If you do not think you will be able to meet this date please let me know immediately.

Kind regards,

Lianne Parkhouse
Royal Society Open Science
openscience@royalsociety.org

on behalf of the Associate Editor, and Professor Mark Chaplain (Subject Editor)
openscience@royalsociety.org

Reviewer comments to Author:

Reviewer: 1

Comments to the Author(s)

- 1) You should give FPR in full and give a brief definition when first introducing it. (I appreciate that this is a reply to a comment on a paper of yours. However, not all readers will have the previous material fresh in their minds.)
- 2) Your use of likelihood ratio is non-standard. This is typically used to mean the ratio of the maximum likelihood under H_1 to that under H_0 . Not the ratio of the integrated likelihood.
- 3) I was puzzled at to why the symbol L_{10} was chosen. Why 10 I wondered. Then it occurred to me that it is 1 for H_1 and 0 for H_0 . I then realised that Johnson had used BF_{10} for the Bayes Factor. I think that it would be best to be explicit..
- 4) The agreement with Johnson is hardly surprising since that also assumes a lump of probability on the null hypothesis. You seem however, to be claiming it as some sort of truly independent validation. (Of course it validates the algebra at least to some degree.) I think you ought to recognise this somewhere.

Reviewer: 2

Comments to the Author(s)

I mostly agree with this response and I believe it deserves to be published. I only want to add a few references that might improve the manuscript. I leave it to the author's discretion whether to include those.

Berger & Sellke (1987, JASA) discuss in Section 4.2 that deviations from point null hypotheses to small sets about H_0 basically give the same results on the corresponding Bayes factors and likelihood ratios. This is reviewed in more detail in Held and Ott (2018), Section 5.2, and would perhaps add to the current discussion.

From a scientific perspective, a related discussion of the suitability of point null hypotheses has appeared in response to the paper by Amrhein and Greenland ("Retire Statistical Significance", 2019, Nature). In particular the letters by Ioannidis ("Retiring statistical significance would give bias a free pass"), Johnson ("Raise the bar rather than retire significance") and Haaf et al ("Retire significance, but still test hypotheses") seem to be relevant in this context. I think the response would be even stronger if it would review and comment on this discussion.

Author's Response to Decision Letter for (RSOS-190819.R0)

See Appendix A.

Decision letter (RSOS-190819.R1)

10-Oct-2019

Dear Dr Colquhoun,

I am pleased to inform you that your manuscript entitled "A response to O. Arandjelovic's critique of "The reproducibility of research and the misinterpretation of p -values"" is now accepted for publication in Royal Society Open Science.

Kind regards,
Lianne Parkhouse
Royal Society Open Science
openscience@royalsociety.org

on behalf of the Associate Editor, and Professor Mark Chaplain (Subject Editor)
openscience@royalsociety.org

Author's response to reviewers

I'm very grateful to the reviewers for their suggestions which have given me the opportunity to improve my paper. Here are my responses (in red) to their comments, and below that is a copy of my paper with tracking on to make it obvious what changes I've made in the revision.

David Colquhoun

Reviewer comments to Author:

Reviewer: 1

Comments to the Author(s)

1) You should give FPR in full and give a brief definition when first introducing it. (I appreciate that this is a reply to a comment on a paper of yours. However, not all readers will have the previous material fresh in their minds.)

Good point. I added an extra paragraph at the start in an attempt to explain briefly.

2) Your use of likelihood ratio is non-standard. This is typically used to mean the ratio of the maximum likelihood under H_1 to that under H_0 . Not the ratio of the integrated likelihood. I guess the problem is that there are a lot of ways to calculate the likelihood ratio. Many of them have been collected in Table 3 of Held & Ott (2108). I think that it's fair to say the value when you've observed $p = 0.05$ is "around 3". I've inserted a reference to this now.

In my 2014 paper, when you've observed $p=0.05$ I calculated a likelihood ratio of 2.76 in favour of H_1 when you know the SD. When you use observed SD and effect size in the simulations (section 10) , it came to 3.6. Both are of the order of 3 and in practice it would make little difference which you used. The Berger & Sellke 2001 value, $1/[-e \ln(p)]$, gives 2.46, also of the order of 3.

3) I was puzzled at to why the symbol L_{10} was chosen. Why 10 I wondered. Then it occurred to me that it is 1 for H_1 and 0 for H_0 . I then realised that Johnson had used BF_{10} for the Bayes Factor. I think that it would be best to be explicit..

Fixed, Held also uses this notation, which I think is helpful.

4) The agreement with Johnson is hardly surprising since that also assumes a lump of probability on the null hypothesis. You seem however, to be claiming it as some sort of truly independent validation. (Of course it validates the algebra at least to some degree.) I think you ought to recognise this somewhere.

Point taken. I added "which test a point null hypothesis" so it now reads " Other approaches which test a point null hypothesis give similar results,"

Reviewer: 2

Comments to the Author(s)

I mostly agree with this response and I believe it deserves to be published. I only want to add a few references that might improve the manuscript. I leave it to the author's discretion whether to include those.

Berger & Sellke (1987, JASA) discuss in Section 4.2 that deviations from point null hypotheses to small sets about H_0 basically give the same results on the corresponding Bayes factors and likelihood ratios. This is reviewed in more detail in Held and Ott (2018), Section 5.2, and would perhaps add to the current discussion.

Both references have been added.

From a scientific perspective, a related discussion of the suitability of point null hypotheses has appeared in response to the paper by Amrhein and Greenland ("Retire Statistical Significance", 2019, Nature). In particular the letters by Ioannidis ("Retiring statistical significance would give bias a free pass"), Johnson ("Raise the bar rather than retire significance") and Haaf et al ("Retire significance, but still test hypotheses") seem to be relevant in this context. I think the response would be even stronger if it would review and comment on this discussion.

I added a reference to Ioannidis and to Wacholder to point out that they use the *p-less-than* approach which underestimates the false positive risk in my opinion. But I don't think that this is the place to discuss the use of $p = 0.005$ as a new bright line. I already discussed that question at length towards the end of section 9 in my 2017 paper.

Now follows the paper with tracking on, to highlight the changes that have been made.

A response to O. Arandjelovic's critique of "The reproducibility of research and the misinterpretation of p -values"

David Colquhoun, UCL

I proposed (8, 1, 3) that p values should be supplemented by an estimate of the false positive risk (FPR). FPR was defined as the probability that, if you claim that there is a real effect on the basis of p value from a single unbiased experiment, that you will be mistaken and the result has occurred by chance. This is a Bayesian quantity and that means that there is an infinitude of ways to calculate it. My choice of a way to estimate FPR was, therefore, arbitrary. I maintain that it's a reasonable way, and has the advantage of being mathematically simpler than other proposals and easier to understand than other methods. This might make it more easily accepted by users. As always, not every statistician agrees. This paper is a response to a critique of my 2017 paper (1) by Arandjelovic (2).

First some minor matters.

In his critique of my 2017 paper (1), Arandjelovic says (2) that my argument "provides little if any justification for the continued use of the p -value". I agree because I never contended that it did. I recommended (3) that authors should

"In my proposal, the terms "significant" and "nonsignificant" would not be used at all. This change has been suggested by many other people, but these suggestions have had little or no effect on practice. A p -value and confidence interval would still be stated but they would be supplemented by a single extra number that gives a better measure of the strength of the evidence than is provided by the p -value"

Furthermore calculation of the p value is the simplest way to calculate what I believe really matters, the false positive risk. It is an input to our web calculator.(4).

He also says (2) "Although criticisms [of p values] are not new, until recently they were largely confined to the niche of the statistical community.". Actually criticisms have been voiced by users for a long time as exemplified by the opening quotation in my paper. Bakan, writing in the *Psychological Bulletin* in 1966 (5) described the conventional null-hypothesis testing as an emperor who had no clothes. I would agree that these early warnings have had little effect on practice, but the reason for that is not primarily a statistical one, but rather a consequence of the perverse incentives that are imposed on both authors and journal editors to produce discoveries without worrying too much about whether the results are just chance.

The major criticism

The main criticism of my piece in ref (2) seems to be that my calculations rely on testing a point null hypothesis, i.e. the hypothesis that the true effect size is zero. He objects to my contention that the true effect size can be zero, “just give the same pill to both groups”, on the grounds that two pills can’t be exactly identical. He then says “I understand that this criticism may come across as frivolous semantic pedantry of no practical consequence: “of course that the author meant to say ‘pills with the same contents’ as everybody would have understood”. Yes, that is precisely how it comes across to me. I shall try to explain in more detail why I think that this criticism has little substance.

Nevertheless I’m grateful to Arandjelovic (2) because his invective has given me the chance to discuss in a bit more detail the assumptions behind my views. Many of his criticisms have already been answered in my 2019 paper (3), which has been online in arXiv for a year since 2018.

One puzzling aspect of his criticisms is that they don’t mention the crucial matter of likelihood ratios.

Likelihood ratios

In my 2017 paper (1), I suggested three different ways of expressing uncertainty that improved on giving p values alone. One was to use the reverse Bayes approach to calculate the prior probability that you’d need to believe in order to achieve a specified FPR. This overcomes the fact that that we usually have no information about the prior probability. But it has the disadvantage that it risks creating a “bright line” at FPR = 0.05, and that sort of dichotomisation is universally deplored (6). It also has the disadvantage that the idea of a prior probability is slippery and unfamiliar to most users.

As an alternative approach, I suggested giving the likelihood ratio, or its equivalent FPR based on the assumption that the prior probability is 0.5. This is now my first choice (3). The ~~likelihood ratio (or Bayes’ factor)~~ is the bit of Bayes’ theorem that represents the strength of the evidence from the experiment, so it is the most natural thing to use (7). Under my assumptions (3) the Bayes’ factor becomes the likelihood ratio. It measures how probable your observations would be under the hypothesis that there is a real effect, relative to how probable they would be under the null hypothesis of zero effect. Call this ratio L_{10} (the subscript indicates that the ratio is defined for \$H_1\$, relative to \$H_0\$ ). The biggest value that this can take is that when the alternative hypothesis has its most likely value, the observed mean effect size. It turns out that even this maximum evidence in favour of the alternative does not reject the null hypothesis as often as the traditional p value. It’s well known that if you observe $p = 0.05$, the likelihood ratio in favour of the alternative hypothesis is only around 3 (e.g. ref (7) and section 10 in ref (8)). (There are many ways to define the likelihood ratio: some are summarised in Table 3 of Held & Ott (11) which shows that most methods imply a value of the order of 3.) Odds of 3 to 1 on their being a real

effect are far less than the odds of 19:1 which might, mistakenly, be inferred from $p = 0.05$.

The use of likelihood ratios~~This approach~~ avoids altogether the arguments about Bayes' theorem. It is an entirely frequentist approach, and it is sufficient alone to show that the p value, as it is often mistakenly interpreted, exaggerates the evidence against the null hypothesis.

Notice that the likelihood ratio approach does not assert that the true effect size is exactly zero. It merely says that if L_{10} is small, then the evidence that there is a real effect is weak. From this point of view it's irrelevant that the null hypothesis is that the true effect size is *exactly* zero. In any case, a narrow distribution around zero gives much the same results (18, 11). I agree with Berger & Sellke (1987) when they say "for a large number of problems testing a point null hypothesis is a good approximation to the actual problem" (18). If the likelihood ratio is not sufficiently big in favour of a real effect to conclude that a real effect is unlikely, then one obviously doesn't conclude that the effect size is exactly zero.

It's important to realise that there are ~~two~~many different ways in which the likelihood ratio can be calculated. In most discussions this is ignored, or at least not stated explicitly, and this has led to confusion. One crucial~~The~~ distinction between them is old. Lindley (1957) pointed it out

"In fact, the paradox arises because the significance level argument is based on the area under a curve and the Bayesian argument is based on the ordinate of the curve." — D.V. Lindley (1957) [17, p.190].

This is illustrated in Figure 1 of ref. (1). Astonishingly there seem to be no generally accepted names for the two approaches. I called them the *p-equals* approach and the *p-less-than* approach, and the distinction between them is discussed in detail in section 3 of my 2017 paper (1) so I won't repeat it here. Most people (e.g. 20, 21) still use the latter approach, though the former is clearly appropriate for answering my question: how should you interpret the p value from a single unbiased experiment. For any given p value, the *p-equals* approach gives smaller likelihood ratio, or bigger false positive risk, than the *p-less-than* approach (see Figure 2 in ref (1)). Neglect of this distinction has led some people to underestimate the risk of false positives.

One problem with advocating likelihood ratios as a measure of evidence is that they are unfamiliar to most users, and that there is no generally accepted criterion for how big they must be for you to have reasonable confidence that your results aren't a result of chance alone. That is why I suggested that, rather than specifying the result as L_{10} , the odds ~~it~~ should be given as expressed as a probability

$$\text{FPR}_{50} = 1/(1 + L_{10}).$$

This can be interpreted, using Bayes' theorem, as the FPR when the prior odds are 1, *i.e.* when the prior probability of a real effect is 0.5 (e.g. see equation A6 in ref (3)). The advantage of doing this is that the FPR measures what most people still mistakenly think the p value does. That makes it very easy for non-statisticians to understand. But the disadvantage is that it involves Bayes' theorem and that always means an outbreak of the statistics wars.

Insofar as the FPR_{50} is just a transformation of L_{10} , it is entirely frequentist, but its interpretation as a posterior probability is not.

How much do p values (as commonly misinterpreted) exaggerate the evidence against the null hypothesis?

Much of the argument in this area centres on whether or not p values (as commonly misinterpreted) exaggerate the strength of the evidence against the null hypothesis. Most people think that they do, though the extent of the exaggeration depends on the precise assumptions that you make (e.g. 18, 11).

Everyone agrees that the p value is not the probability that your results occur by chance alone, despite that being the most common interpretation placed on it by users. The fact that so many people believe this misinterpretation suggests that what they want to know is the probability that their results are due to chance alone, so we need a proper definition of that term. In my 2017 paper(1) I define it this way.

“... when a statistical test of significance comes out positive, what the probability is that you have a false positive, *i.e.* there is no real effect and the results have occurred by chance. This probability is defined here as the false positive risk (FPR)”.

One way of looking at the difference between the p value and the FPR is to note that they have different denominators. The numerator for both is the number of false positives in hypothetical replications of the experiment. If the criterion for a positive result is $p < 0.05$, then this will be 5% of all tests in which the null hypothesis is true. The p value is the ratio of this number of false positives to the total number of tests in which the null hypothesis is true (which is not usually known). The FPR is the ratio of the number of false positives to the total number of positive tests, both false positives and true positives. Under realistic conditions, the former denominator is larger so the p value is smaller than the FPR. A numerical example is given at 26:00 in (9).

Another way to look at the difference between p value and FPR is to look at confusion between them as an example of the error of the transposed conditional Colquhoun (10). They measure quite different things so, in principle, they cannot be equal.

If we accept that what we want to know is the FPR, the question arises of how to calculate it, and that's where the problems begin. From a Bayesian point of view, the FPR is the posterior probability that the null hypothesis is true, $P(H_0 \mid \text{data})$. There are differences of opinion about how it should be calculated. Because it is a Bayesian concept there is literally an infinitude of ways in which it can be calculated. Held & Ott (2018) (11) have reviewed the many possibilities. Which way should we choose?

As Senn (12) has pointed out, the disagreements about how to calculate FPR are essentially disagreements between different versions of the Bayesian argument, rather than a disagreement between frequentists and Bayesians. Frequentists have no way to calculate FPR.

Some people (eg Casella & Berger (13)) have argued that putting a lump of prior probability on the null hypothesis gives the null an unfair advantage over other possibilities. This is a matter of opinion. It seems quite fair to me. Casella & Berger (1987) (14) say

“Their main thesis is that the frequentist P-value overstates the evidence against the null hypothesis although the Bayesian posterior probability of the null hypothesis is a more sensible measure.”

“The large posterior probability of H_0 that Berger and Delampady compute is a result of the large prior probability they assign to H_0 , a prior probability that is much larger than is reasonable for most problems in which point null tests are used.”

“In fact, it is not the case that P-values are too small, but rather that Bayes point null posterior probabilities are much too big!”

“Most researchers would not put a 50% prior probability on H_0 . The purpose of an experiment is often to disprove H_0 and researchers are not performing experiments that they believe, a priori, will fail half the time! We would be surprised if most researchers would place even a 10% prior probability on H_0 .”

Casella and Berger (14) seem to have much more faith in the ability of experimenters to guess the outcome of an experiment than I think is appropriate. Most bright ideas turn out to be wrong so I would guess that a prior probability of 0.5 there being a real effect is often optimistic, rather than being much too low. In my analysis, observation of $p = 0.05$ would imply a prior, $P(H_1)$ of 0.87 to make the FPR the same as the p value (4). They contend that this is reasonable. I think that if you were to submit a paper that claimed you'd made a discovery and that a necessary assumption for that claim to be true was that you were almost (90%) certain that the claimed effect was real before you did the experiment, your paper would be unlikely be accepted. Casella & Berger seem to think that it's legitimate to adjust your prior

in order to make the FPR more or less the same as the p value. This makes no sense at all to me.

I chose to use a point null hypothesis as prior and to use a simple alternative hypothesis ((3), (11)). This makes sense because it's exactly what you do when you simulate repeated t tests, as in Colquhoun (2014) (8). The rest of the prior probability is on the alternative hypothesis, and when this is given its most likely value, the observed mean effect size we find that the likelihood ratio is, at most, about 3, as above.

At the other extreme Senn (12) has shown that a prior can be chosen (for one-sided tests) that makes the p value essentially identical with the FPR. But since the FPR and the p value measure quite different things there is no earthly reason why they should be the same. This seems about as sensible as saying that you can always choose a prior probability that makes the p value the same as the FPR. Without hard evidence about the accuracy of the priors that are assumed these are mere parlour tricks.

The use of a simple alternative hypothesis is not crucial for my results. Other approaches which test a point null hypothesis give similar results, as shown in ref (3). In particular the approaches of Sellke et al, 2001 (15), and of Johnson (2013) (16) give results that are quite close to mine, using priors for the alternative hypothesis that are designed to maximise the odds in favour of rejection of the null hypothesis, H_0 . It turns out that they reject the null hypothesis much less often than the p value. These conclusions strengthen still further the view that the p values, as commonly misinterpreted, exaggerate the strength of evidence against the null hypothesis. The false positive risk, under any realistic assumptions, is bigger than the p value and this must, to some extent, contribute to the reproducibility crisis.

A similar suggestion

Benjamin and Berger (19) in the recent series, *Moving to a World Beyond “ $p < 0.05$ ”* make a suggestion very similar to mine. They suggest that the p value should be supplemented with the maximum Bayes' factor given by their approximation

$$\underline{BF}_{10} \equiv \frac{1}{-e \ln(p)}$$

where e is the base of natural logarithms and p is the observed p value. If you've observed $p = 0.05$ this is 2.46, again of the order of 3, though this method predicts somewhat higher false risks than most others (see Fig 3 and Table 1 in ref 3). For example, if we observed $p = 0.005$, my method implies $FPR_{50} = 0.034$ whereas the Sellke & Berger limit implies 0.067 (Table 1 in ref 3).

Conclusions

It's true that putting a lump of prior probability on the null hypothesis gives a bigger FPR (for any given p value) than many other choices of prior. In that sense my values should perhaps be viewed as maximum FPRs. But in a different sense they are minimum FPRs, because prior probabilities of a real effect may well be lower than the default value of 0.5.

The fact that the Bayesian approach with a lump of prior probability on the null hypothesis gives results that are identical with simply counting false positives in repeated simulations of t tests, is another reason to think that this approach is reasonable (8).

I'd be quite happy for people to report, along with the p value and confidence intervals, the likelihood ratio, L_{10} , that gives the odds in favour of there being a real effect, relative to there being no true effect. That is a frequentist measure and it measures the evidence that's provided by the experiment.

If these odds are expressed as a probability, rather than as odds, we could cite, rather than L_{10} , the corresponding probability $1/(1 + L_{10})$. I suggest that a sensible notation for this probability is FPR_{50} , because it can, in Bayesian context, be interpreted as the false positive risk when you assume a prior probability of 0.5. But since it depends only on the likelihood ratio, there is no necessity to interpret it in that way, and it would save a lot of argument if one didn't.

I think that the question boils down to a choice -do you prefer an 'exact' calculation of something that can't answer your question (the p value), or a rough estimate of something that can answer your question (the false positive risk). I prefer the latter.

References

- (1) Colquhoun D. The reproducibility of research and the misinterpretation of P values. Royal Society Open Science 2017. 4:171085.
<http://dx.doi.org/10.1098/rsos.171085>. Available at
<https://royalsocietypublishing.org/doi/pdf/10.1098/rsos.171085>
- (2) Arandjelovic O. A More Principled Use of the p-Value? Not so Fast: A Critique of Colquhoun's Argument. Royal Society Open Science 2019.
- (3) Colquhoun D. The False Positive Risk: a proposal concerning what to do about p-values. The American Statistician 2019;73(Issue sup. 1: Statistical Inference in the 21st Century: A World Beyond $p < 0.05$):192 -201.
<https://www.tandfonline.com/doi/full/10.1080/00031305.2018.1529622>
<https://www.tandfonline.com/doi/full/10.1080/00031305.2018.1529622#aHR0cHM6Ly93d3cudGFuZGZvbmxpbmUuY29tL2RvaS9wZGYvMTAuMTA4MG8wMDAzMTMwNS4yMDE4LjE1Mjk2MjI/bmVlZEFjY2Vzcz10cnVlQEBA>
 MA== and <https://arxiv.org/ftp/arxiv/papers/1802/1802.04888.pdf>

- (4) Colquhoun D, Longstaff C. False positive risk calculator. 2017. <http://fpr-calc.ucl.ac.uk/>
- (5) Bakan D. The test of significance in psychological research. Psychological Bulletin 1966; 66:423-37. Available at <http://citeseerx.ist.psu.edu/viewdoc/download?doi=10.1.1.441.1136&rep=rep1&type=pdf>
- (6) Amrhein V, Greenland S, McShane B. Scientists rise up against statistical significance. Nature 2019 Mar;567(7748):305-7. <https://www.nature.com/articles/d41586-019-00857-9>
- (7) Goodman SN. Toward evidence-based medical statistics. 2: The Bayes factor. Ann Intern Med 1999 Jun 15;130(12):1005-13.
- (8) Colquhoun D. An investigation of the false discovery rate and the misinterpretation of p-values. R Soc Open Sci 2014 Nov 19;1(3):140216. <https://royalsocietypublishing.org/doi/pdf/10.1098/rsos.140216>
- (9) Colquhoun D. The false positive risk: a proposal concerning what to do about p-values (version 2). 19-1-2018. YouTube, https://www.youtube.com/watch?time_continue=1640&v=jZWgijUnxl
- (10) Colquhoun D. The problem with p-values. Aeon Magazine 2016 Aug 11. <https://aeon.co/essays/it-s-time-for-science-to-abandon-the-term-statistically-significant>
- (11) Held L, Ott M. On p-Values and Bayes Factors. Annual Review of Statistics and Its Application 2018;5:6-1-6-27.
- (12) Senn SJ. Double Jeopardy?: Judge Jeffreys Upholds the Law (sequel to the pathetic P-value). 2015. Error Statistics Philosophy. 2017 <https://errorstatistics.com/2015/05/09/stephen-senn-double-jeopardy-judge-jeffreys-upholds-the-law-guest-post/>
- (13) Casella G, Berger RL. Reconciling Bayesian and Frequentist Evidence in the One-Sided Testing Problem. Journal of the American Statistical Association 1987;82:106-11.

- (14) Casella G, Berger RL. [Testing Precise Hypotheses]: Comment. *Statistical Science* 1987;2(3):344-7.
- (15) Sellke T, Bayarri MJ, Berger JO. Calibration of p values for testing precise null hypotheses. *American Statistician* 2001 Feb;55(1):62-71.
- (16) Johnson VE. UNIFORMLY MOST POWERFUL BAYESIAN TESTS. *Annals of Statistics* 2013;41(4):1716-41.
- (17) Lindley, DV. A statistical paradox. *Biometrika*, 1957; 44, 187-192
- (18) Berger, JO, Sellke,T Testing a Point Null Hypothesis: The Irreconcilability of P Values and Evidence. *Journal of the American Statistical Association*, 1987, 82, 112 – 139
- (19) Benjamin DJ, Berger, JO. Three Recommendations for Improving the Use of *p*-Values. *The American Statistician* 2019;73 (Issue sup. 1: Statistical Inference in the 21st Century: A World Beyond $p < 0.05$):186 -191. Available at <https://www.tandfonline.com/doi/full/10.1080/00031305.2018.1543135>
- (20) Ioannidis JP. 2005 Why most published research findings are false. *PLoS Med*.2,e124.(doi:10.1371/journal.pmed.0020124)
- (21) Wacholder S, Chanock S, Garcia-Closas M, El GL, Rothman N (2004) Assessing the probability that a positive report is false: an approach for molecular epidemiology studies. *Journal of the National Cancer Institute*,96: 434-442.